# From Tumor Mutational Burden to Blood T Cell Receptor: Looking for the Best Predictive Biomarker in Lung Cancer Treated with Immunotherapy

**DOI:** 10.3390/cancers12102974

**Published:** 2020-10-14

**Authors:** Andrea Sesma, Julián Pardo, Mara Cruellas, Eva M. Gálvez, Marta Gascón, Dolores Isla, Luis Martínez-Lostao, Maitane Ocáriz, José Ramón Paño, Elisa Quílez, Ariel Ramírez, Irene Torres-Ramón, Alfonso Yubero, María Zapata, Rodrigo Lastra

**Affiliations:** 1Medical Oncology Department, University Hospital Lozano Blesa, 50009 Zaragoza, Spain; mcruellas@vhio.net (M.C.); mgasconr@salud.aragon.es (M.G.); disla@salud.aragon.es (D.I.); m.ocarizdiez@salud.aragon.es (M.O.); equilezb@salud.aragon.es (E.Q.); mitorres@salud.aragon.es (I.T.-R.); ayuberoe@salud.aragon.es (A.Y.); mzapatag@salud.aragon.es (M.Z.); rlastrad@salud.aragon.es (R.L.); 2Aragon Health Research Institute (IIS Aragón), 50009 Zaragoza, Spain; pardojim@unizar.es (J.P.); lmartinezlos@salud.aragon.es (L.M.-L.); joserrapa@salud.aragon.es (J.R.P.); 3ARAID Foundation (IIS Aragón), 50009 Zaragoza, Spain; 4Microbiology, Preventive Medicine and Public Health Department, Medicine, University of Zaragoza, 50009 Zaragoza, Spain; 5Biomedical Research Center in Bioengineering, Biomaterials and Nanomedicine Network (CIBER-BBN), 28029 Madrid, Spain; 6Instituto de Carboquímica (ICB-CSIC), Miguel Luesma 4, 50018 Zaragoza, Spain; eva@icb.csic.es; 7Immunology Department, University Hospital Lozano Blesa, 50009 Zaragoza, Spain; 8Department of Microbiology, Pediatrics, Radiology and Public Health, University of Zaragoza, 50009 Zaragoza, Spain; 9Aragon Nanoscience Institute, 50018 Zaragoza, Spain; 10Aragon Materials Science Institute, 50009 Zaragoza, Spain; 11Infectious Disease Department, University Hospital Lozano Blesa, 50009 Zaragoza, Spain; 12Nanotoxicology and Immunotoxicology Unit (IIS Aragón), 50009 Zaragoza, Spain; aramirez@iisaragon.es

**Keywords:** lung cancer, ICIs (immune checkpoint inhibitors), biomarker, TMB (tumor mutational burden), TCR (T cell receptor), TCRβ (TCRB), neoantigen

## Abstract

**Simple Summary:**

Immune control inhibitor drugs (anti-PD1/PD-L1/CTLA-4) (ICIs) are showing efficacy in the treatment of lung cancer. Currently the only biomarker with clinical utility for ICIs, such as the expression of PDL1, does not appear to be perfect or effective. Our working group is conducting a pilot study in lung cancer patients receiving ICIs with the aim of analyze the factors that affect the sensitivity of the immunotherapy in lung Cancer. Tumor Mutational Burden (TMB) and the sequencing of the T Cell Receptor (TCR) repertoire in peripheral blood have been postulated as predictive biomarkers of efficacy of immunotherapy. The review focusses on the predictive value of TMB for clinical responses to ICIs and discusses its clinical need after a discussion of the limitations. TCR CDR3 beta analysis and parameters such as clonality and TCR convergence may be good alternatives. For the moment, the combination of biomarkers may be the optimal tool.

**Abstract:**

Despite therapeutic advances, lung cancer (LC) is one of the leading causes of cancer morbidity and mortality worldwide. Recently, the treatment of advanced LC has experienced important changes in survival benefit due to immune checkpoint inhibitors (ICIs). However, overall response rates (ORR) remain low in unselected patients and a large proportion of patients undergo disease progression in the first weeks of treatment. Therefore, there is a need of biomarkers to identify patients who will benefit from ICIs. The programmed cell death ligand 1 (PD-L1) expression has been the first biomarker developed. However, its use as a robust predictive biomarker has been limited due to the variability of techniques used, with different antibodies and thresholds. In this context, tumor mutational burden (TMB) has emerged as an additional powerful biomarker based on the observation of successful response to ICIs in solid tumors with high TMB. TMB can be defined as the total number of nonsynonymous mutations per DNA megabases being a mechanism generating neoantigens conditioning the tumor immunogenicity and response to ICIs. However, the latest data provide conflicting results regarding its role as a biomarker. Moreover, considering the results of the recent data, the use of peripheral blood T cell receptor (TCR) repertoire could be a new predictive biomarker. This review summarises recent findings describing the clinical utility of TMB and TCRβ (TCRB) and concludes that immune, neontigen, and checkpoint targeted variables are required in combination for accurately identifying patients who most likely will benefit of ICIs.

## 1. Introduction

Immunotherapy has become a powerful therapeutic weapon against cancer in recent years, achieving long-lasting responses and significant survival benefits in multiple types of tumors. Thus, anti-programmed cell death-1/programmed cell death-ligand 1 (PD-1/PD-L1) antibody has been approved for second-line or first-line treatment in melanoma, renal cell carcinoma, head and neck squamous and gastroesophageal cancer [1].

In the setting of non-small cell lung cancer (NSCLC) immunotherapy has become a standard of treatment for previously untreated advanced NSCLC without driven mutations. These new treatments include the combination of immune checkpoint inhibitors (ICIs) with platinum-based chemotherapy or ICIs alone [2].

Despite the current success of immunotherapy, not all patients respond similarly and of those responding, serious toxic effects are sometimes observed. Therefore, concern arises on how to identify biomarkers that allow us to recognize immunogenic tumors that will benefit most from ICIs.

Research in the field of biomarkers in immunotherapy aims to characterize the relationship between tumor, immune system, microenvironment, and host.

PD-L1 expression as a predictive biomarker across different types of tumors has been evaluated in some clinical trials but its use as a robust predictive biomarker has been confounded with a number of biological and technological variables. Therefore, there is a need for biomarkers for better stratification [2].

Neoantigens are antigens that arise from altered peptides formed as a result of tumor mutations or encoded by viral proteins. The relevance of neoantigens in cancer treatment has been revealed with the advent of immunotherapy, whose efficacy lies in the existence of such neoantigens potentially recognised by host T cells [3]. Neoantigens are generated as a consequence of somatic mutations [4] (most of them located in exons such a single nucleotide mutations), insertions, or deletions in the coding regions of proteins, chromosomal translocations, post-translational modifications, and alternative splicing [5]. First, the mutated sequence must be expressed by the tumor cell. Then it has to be processed and presented by the patient’s major histocompatibility complex (MHC) molecules and recognized by a T cell receptor (TCR). The probability of having neoantigen-specific T lymphocytes correlates directly with the mutational load [6].

From the point of view of immunotherapy, patients with the highest number of clonal neoantigens are the most likely to respond to treatment. Therefore, tumor neontigens constitute a predictive biomarker model for the response to ICIs and tumor mutational burden (TMB) could be used to indirectly assess the neontigens load [7].

In addition to TMB and PD-L1, other biomarkers also seem to affect to response to immunotherapy in cancer [8,9,10]. Gene expression signatures (IFN-γ signaling and activated T-cells associated with improved OS with second-line atezolizumab in advanced NSCLC [11], tumor-specific genotypes (such as genomic alterations in EGFR, ALK, KRAS, etc., and what is more, their interrelatedness with TMB) have been correlated to immunotherapy response.

Other potential predictor of response is the density of tumor-infiltrating lymphocytes (TILs) in the tumor microenvironment. This reflects a more inmune recognition of tumor cells and leads to an inflamed tumor phenotype which is more sensitive to imnunotherapy. Differents studies have observated high levels of TILs with better response to checkpoint blockade and improved survival across differents types of tumors including NSCLC [9].

Serum-based markers are being investigated and although they need prospective validation, they could be an attractive option especially when the tumor sample is insufficient for testing [9]. High neutrophilto-lymphocyte ratio (NLR) enhances angiogenesis and tumorigenesis and may be a negative prognostic indicator of response to immunotherapy in patients with cancer including metastatic NSCLC [12]. Other peripheral serum markers are the absolute eosinophil count, the absolute monocyte account, and the platelet-to-lymphocyte count [1].

In the current article, the authors review the biomarker TMB as well as other emerging and investigational markers that have potential to better predict responders to immunotherapy in lung cancer.

## 2. TMB and Lung Cancer

### 2.1. What is TMB and How to Measure?

TMB can be defined as the total number of nonsynonymous mutations per megabase of genome examined, being a mechanism generating neoantigens and conditioning the tumor immunogenicity. The presence of non-synonymous somatic mutations can lead to modifications in the encoded proteins that can be recognized by the immune system as non-self and become a neoantigen capable of activating the antitumor immune response.

Consequently, tumors with high mutational load could be elegible to successful treatment with ICIs [2]. However, of all the modified proteins, only a small fraction result in neontigens. It may be the case that a tumor has low mutational load but that these mutations have resulted in neoantigens capable of generating a strong antitumor T response.

The number of somatic mutations varies greatly between different tumors. NSCLC has one of the highest mutation frequencies (0.1 to 100 mut/Mb) [13].

The main pathways contributing to mutation rate are DNA replication pathways and DNA damage repair [14]. Mutations in DNA replication genes, such as POLD1 and POLE, are associated with increased mutation frequency and high TMB [15]. Mutations in mismatch repair system genes (MMR) results in microsatellite inestability (MSI) and high TMB. Pembrolizumab received accelerated approval in the United States by the Food and Drug Administration (FDA) in May 2017 for the treatment of adult and pediatric patients with unresectable or metastatic solid tumors dMMR that progress to standard treatment and have no satisfactory results or alternative treatment options. regardless of the histological subtype or origin of the primary tumor. It was based on data from the KEYNOTE-164 and from the KEYNOTE-158 study. This marked the first approval of an oncology therapy independent of tumor histology or anatomical location of origin, in which treatment is based on a common tumor biomarker.

Recently, the FDA has accelerated the approval of pembrolizumab for the treatment of adult and pediatric patients with unresectable or metastatic cancer with high TMB (≥10 mutations/megabase) that have progressed prior treatment and who have no satisfactory alternative treatment options based on a retrospective analysis of 10 cohorts of patients enrolled in KEYNOTE-158 trial [16]. This approval strengthens the role of genomics in the treatment of cancer.

### 2.2. Clinical Features of TMB

Different studies have investigated whether there is an association between TMB and clinical characteristics in lung cancer (LC). TMB seems to be correlated with gender, being higher in men than in women [17].

Based on the histology of the tumor, many studies conclude that squamous histology lung carcinoma have a higher TMB than those with adenocarcinoma or small cell lung cancer (SCLC). Degree of tumor differentiation and tumor stage have also been described as predictive factors of TMB [18].

NSCLC genomic profile differs significantly between smoking and non-smoking patients. TMB can be influenced by tobacco use due to the accumulation of somatic mutations caused by carcinogens in tobacco smoke which would lead to a higher neoantigen load [19]. This would lead to the idea that patients with tumors harboring smoking signature respond better to ICIs. Some patterns of mutational changes found in smokers like dominant C > A mutations are associated with clinical benefit [20]. Deleterios mutations in POLD1, POLE, and MSH2 contribute to high TMB and they are more frequent in responding smoking patients compared to non-responders [20].

Mutations in KRAS and STK11 [21] are related to smoking, while those of EGFR [22], METDex14, and rearrangements of ALK, ROS1, RET, and NTRK [23] are associated with non-smokers. ALK, ROS1, EGFR, BRAFV600E, and METDex14 mutations are associated with low TMB, while mutations in KRAS/STK11, BRAFnon-V600E are associated with high TMB. This may explain the lower efficacy of ICIs in non-smoking LC [24].

In addition, some specific alterations in patients with advanced NSCLC such as amplifications in MDM2 and MDM4 have been associated with hyperprogression and alterations in B2M and JAK2 have been described as mediating acquired resistance mediated through defective interferon gamma signaling [25].

### 2.3. Measurement of TMB in Tumor Tissue (tTMB)

Determination of TMB is possible thanks to the development of next generation sequencing (NGS) technologies. TMB in tumors is usually measured with whole exome sequencing (WES) or comprehensive genomic profiling (CGP).

WES is the gold standard method for assessing TMB. WES allows the detection of somatic mutations presents within the entire exome (2% of the human genoma, i.e., 30 to 50 Mb of coding sequences) at a lower cost than whole genoma sequencing [26].

Targeted gene panel sequencing is an alternative option to TMB estimation because of its reduced costs and turnaround time required. This method allows an analysis of selected “driver” genes using less amounts of DNA and improving mutation detection sensitivity [26].

There is a high correlation between TMB assessed by targeted gene panel with TMB analysis by WES [25,27].

### 2.4. Blood-Based Tumor Mutation Burden (bTMB)

Although tumor tissue samples are considered the standard material for performing NGS analyzes, up to 30% of patients with NSCLC do not have adequate tissue available at diagnosis. The use of liquid biopsy containing circulating tumoral DNA (ctDNA) instead of tissue represents an attractive alternative for these patients [4].

It offers some advantages due to its noninvasive nature, ability to capture tumor heterogeneity and it allows monitoring treatment response and assessment of residual disease. Moreover, it is less expensive, and it requires short processing time with low failure rate.

bTMB seems to be a good predictive biomarker of clinical benefit from ICIs, although the correlation with tTMB determination is not clear yet [24,26]. bTMB does not include insertions and deletions and there are some differences in sample characteristics such as the source of DNA, collection time, sample type, stage at diagnosis, tissue purity, the ctDNA, maximum somatic allele frequency (MSAF) and the cell-free DNA (cfDNA) input mass.

Moreover the treatment with DNA damaging agents, such as chemotherapy and the different cut-off to characterize bTMB are many factors that could explain the absence of a higher correlation between these parameters [27].

## 3. TMB, From Great Expectation

### 3.1. Studies as Predictive Factor (Table)

To date, the only FDA-approved predictive biomarkers to ICIs in NSCLC is PD-L1 expression [28].

However, many clinical trials show correlation between high TMB and greater response to ICIs across a wide array of tumor histologies [29]. The most robust responses to ICIs have been seen in microsatellite instability (MSI) colorectal cancer, melanoma and NSCLC, which are both tumors with high TMB [30].

In Table 1 we summarize the main studies on TMB analysis as a biomarker of response to ICIs [25,31,32,33,34,35,36,37,38,39].

### 3.2. Strengths

There are several data supporting the use of TMB as a biomarker for ICI efficacy:

**TMB is an effective and independent predictive biomarker of PD-L1 IHC expression in tumors** [41,42,43]. However a greater benefit has been observed with ICIs in patients with high expression of TMB and PD-L1, which suggests that a composite of both may be most helpful in identifying with precision patients most likely to benefit to ICIs [44,45]. This finding is supported by the hypothesis that patients with high PDL1 expression and high TMB are predicted to contain higher frequencies of primed antitumor T cells which are unfunctional due to PD1-mediated inhibition.

**TMB can be efficiently assessed using targeted sequencing panels** such as the Memorial Sloan Kettering-Integrated Mutation Profiling of Actionable Cancer Targets (MSK-IMPACT) and FoundationOne CDx™ (Foundation Medicine, Cambridge, USA) which have been developed and approved by the FDA [46,47]. These platforms have been shown strong correlation with WES and offer the advantage of being able to determine the TMB at the same time as druggable mutations [25,48].

**bTMB can replace the tTMB as a predictor of benefit to immunotherapy.** In the randomized POPLAR and OAK clinical trials, a correlation between bTMB and tTMB was observed [37].

The clinical validation of bTMB was reported in a phase III clinical trial, BFAST (Blood-First Assay Screening Trial), evaluating the efficacy of multiple targeted therapies for patients with advanced NSCLC. A positive correlation between tTMB and bTMB scores was observed [37]. The interim results for the phase II B-F1RST study, analyzing bTMB for stratification of atezolizumab in first line therapy NSCLC; bTMB correlated well with treatment response [40].

A high TMB may be a useful biomarker for assessing patient’s risk of inmune-related adverse events (irAES) during anti-PD-1 therapy.

ICIs enhances inmune responses which can cause tumor regression and irAEs. It is not clear why immune side effects occur in some patients but not in others, or why similar treatment can cause different toxicity in different patients. In addition, different types of irAEs could indicate an immunotherapeutic response in different tumor types.

In LC, early development of immune toxicity and low-grade toxicity were correlated with a better response to immunotherapy and a survival benefit as well as endocrine toxicity [49,50,51,52] Regarding skin toxicity, the data are controversial. Since skin irAEs include various types of skin disorders, the association of each skin irAE with outcome may vary [53].

It has been proposed that the association between an improved response to anti-PD-1 therapy and irAES are linked via an underlying neoantigenic potential derived from a high TMB [54]. Last studies suggest a positive correlation between the reporting odds ratios (RORs) of presenting an irAE during anti-PD-1 therapy and the corresponding TMB across multiple cancer types [55]. In this way, cancers with a high TMB, such as melanoma and NSCLC, are associated with a higher irAE RORs during anti-PD-1 therapy than cancers with a low TMB.

In addition to TMB, other biomarkers such as T cell diversity, cytokines, inflammatory factors, and gut microbiome may be useful biomarkers for assessing patients’ risk of irAEs during anti-PD-1 therapy [1].

## 4. To Important Doubts

### 4.1. Limitations

Despite the promising results indicated above, several evidences highlight the potential limitations of using TMB as a predictive biomarker:

**Technical limitations regarding TMB analysis**: Long turnaround time: tissue TMB analysis takes 2–3 weeks and storage time: the mean mutation number decreased as sample storage time increased. In contrast to the assessment of PD-L1 expression; TMB analysis requires a significant amount of tissue [56]. Moreover, tumor tissue biopsies used for TMB analysis are often fixed with formaldehyde which can induce crosslinks that are a main source of sequencing artifacts [26].

**Absence of standardization of TMB.** The established cut-off values differ according to the type of tumor studied, controlled sequencing technique, region, and size sequenced as well as the type of mutation included in the TMB analysis.

One WES study defined the cut-off point that combined maximal sensitivity (100%) with maximal specificity (67%) at 178 non-synonymous mutations in advanced NSCLC patients [31]. Using sequencing panels, the cut-off points are set to values of 7 to 10 mut/Mpb for LC [35].

**TMB does not seem to correlate properly and linearly with the neontigen load.** Although there is a correlation between TMB and neoantigen load [31] a large number of mutations does not always translate into greater neoantigens generation [42]. Thus, there are patients with high TMB who does not respond to ICIs and others with low TMB who have more durable responses. Therefore, additional factors to TMB may contribute to response to ICIs such as intratumor heterogeneity, gene expression signature, hydrophobicity, or tumor microenvironment [43,44,45].

Moreover, patients with fewer but more relevant mutations, such as alterations in the MMR pathway and homologous recombination pathway (HR), might respond better to ICIs than patients who have a higher TMB but with fewer mutations in certain crucial loci [57]. It should be noted that mutated proteins should produce adequate peptides containing the neoantigen epitopes to be presented by MHC-I and MHC-II which is required to activate CD4+T cells and subsequently CD8+T cells. Moreover, it should be taken into account that some MHC haplotypes might present better neoantigen peptides to cells than others and thus, patients expressing those haplotypes could respond better to ICIs.

In addition, Merkel-cell carcinoma and some virus-induced tumors suggest that the presentation of viral antigens on certain tumor types may confer an increased response rate to ICIs [58].

Another limitation of TMB is that it is unable to identify immunogenic tumors sensitive to immunotherapy that have neoantigens other than those derived from non-synonymous mutations [58].

### 4.2. Studies as Negative Predictive Factor

Despite the strong rational of TMB, the latest evidence suggests that the predictive value of TMB as a biomarker of response to ICIs may have limited clinical utility.

Several trials, including KEYNOTE-010 and KEYNOTE-042, have retrospectively demonstrated that TMB could be useful as predictive marker of response to ICIs but it was not a useful tool for predicting response to chemotherapy [48,49]. The same results have been observed in KEYNOTE-021/189 and 407 trials in which no statistically significant association was determined between tTMB and efficacy of pembrolizumab plus chemotherapy or chemotherapy alone as first-line treatment for NSCLC. Taking into account the results of these studies, TMB does not seem to identify responders from non-responders either for the combination treatment or chemotherapy alone.

One of the hypothesis that could explain these results obtained in the previously mentioned studies is the obliteration of the predictive ability of TMB in the context of superimposed cytotoxic effect of the chemotherapy [8].

In addition, in the CHECKMATE-026 study, no differences in OS were found in patients stratified according to the TMB [33].

The data from the final analysis of the CHECKMATE-227 met its primary endpoint of OS in patients with PD-L1 ≥ 1% treated with Nivolumab + Ipilimumab. Based on these results, the European Society for Medical Oncology clinical practice guidelines have proposed the use of TMB to select patients with NSCLC for first-line nivolumab and ipilimumab [8]. However, no consistent correlation was observed between survival outcomes with ICIs vs chemotherapy and PD-L1 or TMB alone or in combination.

Similar to PD-L1 expression, TMB is an imperfect biomarker requiring refinement and future studies are warrented to improve its value.

In Table 2 we summarize the main studies on TMB analysis as a negative biomarker of response to ICIs [33,58,59,60,61,62,63].

## 5. Is the TMB Dead?

TMB is not ready for routine use in clinical practice and TMB must still be validated. The predictive role of TMB is enhanced when combined with the expression of PD-L1. Both biomarkers appear to be complementary. TMB is able to identify a subgroup of patients PD-L1 non-expressers or low expressers that benefit more from ICI treatment [8].

New biomarkers are being assessed as candidate for ICIs efficacy in NSCLC. Among these biomarkers are those related to the tumor microenvironment such as tumor infiltrative lymphocytes (TILs) [28] or interferon signaling, CD8 + PD − 1 + Tcells expressing CD28 [65] and gut microbiome in the feces [66].

The benefit of adding TMB analysis to the previous parameters is controversial. No statistically significant association between TMB and CD8 lymphocytes measured by IHC has been demonstrated in patients with LC treated with pembrolizumab [67] and association between TMB and T-cell inflamed gene expression profile in patients treated after PD-1 axis blockade is controversial [68].

The combination of multiple biomarkers may be the optimal tool to predict ICIs efficacy in NSCL patients as well as the arrival of new biomarkers such as T-cell clonality and restricted TCR-repertoire in the blood and tumor tissue [69].

## 6. New Emerging Biomarkers: TCRB T Cell Receptor Beta (TCRβ)

### 6.1. What is TCRB and How to Measure

TCR are antigen specific receptors which are essential to immune response and are present on the cell surface of T lymphocytes. The clonotypic TCR is a complex formed by a heterodimer (αβ or γδ) joined by disulfide bonds responsible for antigen recognition in the context of MHC and a series of invariant chains: CD3 (γ, δy ε) y CD247 (ζ). Complementarity-determining regions (CDRs) are part of the variable chains in TCR and they are crucial to the diversity of antigen specificities generated by lymphocytes. Among those, the CDR3 shows the greatest variability and is coded by the combination between segments of the VJ regions.

TCR participates in the positive and negative selection of the T repertoire during thymic maturation. Subsequently, on the periphery, it is responsible for the recognition of antigens, and triggers the expansion and differentiation of T cell clones.

Unlike TMB, TCR convergence detects T cell responses to any antigen, including neoantigens beyond those arising from non-synonymous mutations as aberrant post-translational modifications, ectopic gene expression, splicing defects, autoantigens, and virus-derived antigens. In addition, TCR convergence avoids probabilistic models for prediction of immunogenicity; is sequencing efficient, typically requiring less than 2M reads per sample; and may be measured from the abundant genetic material within the buffy coat fraction of centrifuged peripheral blood to enable liquid biopsy applications [70,71].

Recents studies have identified that sequencing by NGS the complementary-determining region 3 (CDR3) from rearranged TCR variable beta (Vβ) (TCRB) chain can be used to evaluate and measure the clonality and diversity of T cells in both peripheral blood and tumor sample [72].

### 6.2. Studies as Predictive Factor

Motivated by the deficiencies of existing non-invasive biomarkers, the use of peripheral blood TCRB repertoire sequencing has emerged as a new predictive biomarker of response to ICIs.

Focusing on the results of the recent studies, it has been postulated that changes in the repertoire of T cells in peripheral blood after ICIs could be a biomarker of response to this treatment [73]. It has been observed that increasing TCR diversity after ipililumab treatment was associated with better outcomes in melanoma patients [74,75]. In fact, studies are currently underway evaluating whether changes in the diversity of TCR in peripheral blood could be correlated with better responses to ICIs.

TCR evenness and TCR convergence could be predictive markers of response to ICIs. TCR evenness is a measure of the similarity of clone frequencies in a TCR repertoire and this term is equivalent to “clonality”. A mature T cell has a unique genetic sequence and different clones with the same TCR will proliferate for a response. TCR convergence is defined as the process whereby antigen-driven selection enriches for T cell receptors having a different genetic sequence but the same resulting amino acid sequence due to codon degeneracy and a shared antigen specificity. TCR convergence can arise in response to a broad range of tumor associated antigens, suggesting that elevated TCR convergence could define the tumor infiltrating T cell repertoire being a feature of the tumour microenvironment as seen in melanoma and NSCLC [76]. This concept has been proposed as an indicator of tumor immunogenicity and thus its sensitivity to ICIs [70,72,75,77].

Reduced T cell evenness and elevated TCR convergence evaluated by The Oncomine TCRB-SR assay which amplifies the CDR3 region of the TCRB chain were identifying as features of the pretreatment NSCLC tumor microenvironment of responders to anti-PD-1 blockade [72,75]. Another analysis of baseline peripheral blood TCRB from individuals receiving CTLA-4 blockade also indicated that convergence and evenness values independently predicted response to immunotherapy and the combination of these features with established biomarkers such as PD-L1 expression derived from transcriptional profiling of the tumor microenvironment improved the accuracy of the response [71,75].

### 6.3. Strengths

TCR-based features could be a better biomarker to immunotherapy treatment than TMB and PD-L1 IHC staining including those cancers in which TMB does not predict ICIs response [71,72]. Thus, it has been seen that convergence values could discriminate between responders and non-responders to treatment with ICIs with significant accuracy, compared to the historical performance of TMB as a biomarker [70].

TCR can be used as a useful tool to establish the development and prognosis in LC patients by dynamically detecting the TCR repertoire during treatment. It has been observed that TCR repertoire differs between healthy controls and LC patients in terms of diversity, CDR3 clonotype, V/J segment usage, and sequence and a high baseline diversity was correlated with better immune status and clinical benefit [78]. This fact is important, because considering that baseline TCR repertoire correlates with certain clinical characteristics, we could identify a subpopulation of patients who would benefit more from ICIs.

### 6.4. Limitations

Care must be taken with the rate of false positives in the detection of TCR convergence, either due to the clustering of functionally different clones, or the presence of artificial clones derived from residual substitution errors [70]. For all these reasons, it is necessary to sequencing platforms that minimize the false positive rate in the detection of convergent TCRs.

Different platforms such as Ion Torrent and Illumina have been used so far to analyze the TCR convergence through the TCRB repertoire sequencing. This attempt has been limited by substitution sequencing errors which can create artefacts resembling TCR convergence. In this context, it has been compared Ion Torrent to Illumina assays with consistent data as measurements of TCR evenness, diversity, and clonal overlap. However, it seems that Ion Torrent may be more suitable to the measurement of TCR convergence than Illumina [70,71].

Future studies will be necessary to clarify the prognostic and predictive value of TCRB convergence as an immune repertoire biomarker.

## 7. Correlation TMB-TCRB

TMB and TCR diversity have been suggested as predictive biomarkers in cancer immunotherapy but it is not clear how TMB or clinical factors correlate with TCR clonality.

Nowadays, there are no published data on the predictive role of TCR clonality and diversity and the correlation with other biomarkers, such as PD-L1 and TMB.

Recently, the possible association between these two biomarkers has been investigated. In total, 43 NSCLC patients and 18 SCLC patients with no TKI-related driver gene mutations were enrolled in this study [79]. TMB was determined with LC tissue by WES and TCR sequencing was performed with peripheral blood samples. It was observed that chemotherapy treatment could decrease TMB and patients with decreased TMB after the first-line chemotherapy could benefit less from immunotherapy. A significant decrease in TMB from mean value of 3.67 to 1.95 was observed after chemotherapy in NSCLC patients (*p* =0.05).

The number of TCR clones was not correlated with TMB, gender, age, metastasis, LC subtype or therapeutic response (*p* > 0.05). However, TCR diversity was higher in patients with a smoking history than those with no smoking history (*p* = 0.01) [79,80].

## 8. Conclusions

LC is today one of the leading causes of morbidity and mortality worldwide. Therapeutic management of metastatic LC is a clinical challenge, therefore the identification of biomarkers that allow us to select which patients will benefit most from treatment with ICIs is key to clinical decision making and personalized therapeutic approach. TMB has emerged recently as a new predictive biomarker for ICIs response in NSCLC, either alone or in combination with PD-L1 expression levels, might separate responders from nonresponders to ICIs. However, this biomarker needs to be validated for routine clinical use and certain doubts arise taking into account the latest studies that show conflicting results on its usefulness. For these reasons, nowadays TMB appears not to be ready for routine use in clinical practice. In this setting, TCR-based features as TCR convergence and evenness has emerged as a novel prognostic and predictive biomarker to response to ICIs. Nevertheless, a combination of biomarkers may be the optimal tool to predict ICIs efficacy in NSCL patients.

The search and application of biological markers that offer reliable information is essential for the development of precision medicine. Therefore, future studies will be necessary to develop a better understanding of molecular LC biology.

Our working group is conducting a pilot study in this direction in LC patients receiving ICIs with the aim of identifying molecular biomarkers that help us predict the efficacy of immunotherapy and establish the most appropriate treatment for our patients.

## Figures and Tables

**Table 1 cancers-12-02974-t001:** Studies on tumor mutational burden (TMB) analysis as a biomarker of response to immune checkpoint inhibitors (ICIs).

Drug Trial	Study Type/Phase	Line of Therapy	Pts, n	Patient Population	Tmb Method & Cutoff	Clinical Outcomes	Author/Year
Pembrolizumab	Retrospective	First line, second or higher	16 of POPLAR trial; 18 of OAK study	Advanced NSCLC	WES: high≥ 178 mutations	TMB was correlated with better ORR (63% vs. 0%, *p* = 0.03), PFS (14.5 vs. 3.7 m, *p* = 0.01) and DCB.	Rizvi NA 2015 [31]
CHECKMATE-026 Nivolumab (NCT02041533)	Exploratory retrospective analysis of phase III study	First line	312	Stage IV or recurrent NSCLC with PD-L1 ≥1%	WES: highTMB ≥243; low TMB <100 mutations	High TMB pts: PFS 9.7 vs. 5.8 m (HR 0.62; 95% CI, 0.38 to 1.00) and ORR (46.8% vs. 28.3%) in nivolumab group compared to chemotherapy.	Carbone D,2017 [33]
CHECKMATE-012 Nivolumab& ipilimumab (NCT01454102)	Phase I	First line	75	Advanced NSCLC	WES: high TMB > median, 158 mutations; low TMB ≤ median	ORR, DCB, PFS were superior in pts with high TMB vs. low TMB (ORR 51% vs. 13%, *p* = 0.0005; DCB 65% vs. 34%, *p* = 0.011; PFS HR 0.41).	Hellmann MD 2018 [34]
CHECKMATE-227 Nivolumab + ipilimumab (NCT02477826)	Phase III	First line	299	Stage IV or recurrent NSCLC	FoundationOne CDx assay; high TMB: ≥10 mut/MbV	PFS was longer among pts with high TMB (mPFS: 7.2 vs. 5.5 months, HR 0.58, *p* < 0.001) in nivolumab + ipilimumab group compared to chemotherapy	Hellman MD 2018 [35]
CHECKMATE-568 Nivolumab + ipilimumab (NCT02659059)	Phase II	First line	288	Stage IV NSCLC	FoundationOne CDx assay; high TMB: ≥10 mut/Mb	ORR was higher (>40%) in high TMB	Ramalingam SS 2018 [32]
CHECKMATE-032 Nivolumab ± ipilimumab (NCT01928394)	Exploratory	Second-line or higher	211	Advanced SCLC	WES: TMB was grouped by tertiles: low, 0 to <143; medium, 143 to 247; high, ≥248 mutations	ORR: 46.2% vs.16%; 1-year PFS: 30% vs. 6.2% 1-year OS: 62.4% vs. 23.4% was higher in pts with TMB high vs TMB low	Hellmann MD 2018 [36]
PD-1 or PD-L1 inhibitors	Retrospective	First line, second or higher	240	Advanced NSCLC	MSK-IMPACT TMB was grouped by percentiles: high TMB >50%	More disease control (complete/partial response vs stable/progression disease) and longer PFS for patients with high TMB >50%	Rizvi H, 2018 [25]
POPLAR & OAK Atezolizumab	Retrospective	Second-line or higher	211 (discovery cohort with 16 p) in POPLAR trial, 583 (validating cohort with 18 p) in OAK study	Advanced NSCLC	Foundation One; bTMB: High bTMB ≥16; low TMB ≤16.	High bTMB (≥16 mut/Mb) was associated with improved PFS, ORR and duration of response.	Gandara DR, 2018 [37]
LACE-BIO-2 Adjuvant Cisplatin (NCT01294280)	Retrospective	Adjuvant chemotherapy	>900	Early-stage NSCLC	Targeted NGS panel using Illumina HiSeq 2000. TMB was categorized into tertiles (low, ≤4 mutations/Mb; intermediate, >4 and ≤8 mutations/Mb; high, >8 mutations/Mb)	High TMB (>8 mut/Mb) was prognostic for favorable OS, PFS, LCSS in patients with resected NSCLC. LCSS benefit with adjuvant chemotherapy was more pronounced in low TMBs (≤4 mut/Mb).	Devarakonda S, 2018 [38]
Neoadjuvant nivolumab	Exploratory	Neoadjuvant PD-1 Blockade	22 (21 were eligible for inclusion)	Surgically resectable early (stage I, II, or IIIA) NSCLC.	WES: highTMB: 311 ± 55 media vs low TMB:74 ± 60 mean	In pts with high TMB (sequence alterations; mean, 311 ± 55 vs. 74 ± 60, *p* = 0.01) a major pathological response was observed.	Forde PM, 2018 [39]
B-F1RST Atezolizumab (NCT02848651)	Phase II	First line	152 (119 were included in the biomarker evaluable population)	Locally advanced or metastatic NSCLC	Foundation Medicine panel; bTMB: high bTMB ≥ 16, versus low bTMB ≤ 16	It was observed a relationship between increasing bTMB score and improved clinical outcomes. ORR and PFS were superior in pts with high bTMB vs low bTMB: ORR 28.6% vs. 4.4%; PFS 4.6 months vs. 3.7 months, HR 0.66 (90% CI 0.42–1.02).	Velcheti V, 2018 [40]

**Table 2 cancers-12-02974-t002:** Studies on TMB analysis as a negative biomarker of response to ICIs.

Drug Trial	Study Type	Pts, n	Patient Population	Purpose of Study	Tmb Method & Cutoff	Clinical Outcomes	Conclusion
KEYNOTE-010 (NCT01905657)	Exploratory retrospective analysis of a randomised controlled trial phase II/III	253 (24% from the all sample)	Previously treated or untreated advanced NSCLC PD-L1(+) with tumour proportion score (TPS)≥ 1% having evaluable Ttmb	Association between tTMB and clinical benefit with pembrolizumab monotherapy	tTMB determined by WES of tumour and matched normal DNA Cutpoint of 175 mutations per exome	tTMB ≥ 175: OS 14.1 m vs. 7.6 m (CI, 0.38–0.83); PFS 4.2 m vs. 2.4 m (CI, 0.40–0.87); ORR 23.5% vs. 9.8% with pembrolizumab and chemotherapy respectively	tTMB was associated with OS, PFS and ORR for the pembrolizumab arms but tTMB was not associated with outcomes for chemotherapy [59,60].
KEYNOTE-042 (NCT02220894)	Exploratory retrospective analysis of a randomised controlled trial phase III	793 (62% from the all sample)				tTMB ≥ 175: OS 21.9 m vs. 11.6 m (CI, 0.48–0.80); PFS 6.3 m vs. 6.5 m (CI, 0.59–0.95); ORR 34.4% vs. 30.9% with pembrolizumab and chemotherapy respectively	
KEYNOTE-021 (NCT02039674)	Exploratory analysis of a randomised controlled trial phase I/II study	267 (48% of patients in cohorts C and G)	Stage IIIb/IV non-squamous NSCLC	Association of tTMB with outcomes for pembrolizumab + chemotherapy and for chemotherapy	tTMB determined by WES of tumour and matched normal DNA Cutpoint of 175 mutations per exome	In cohort G, ORR was higher with pembrolizumab + chemotherapy vs chemotherapy in the 31 patients with tTMB ≥175 mutations per exome (71.4% vs. 30%)	No significant association was determined between tTMB and efficacy of pembrolizumab + chemotherapy or chemotherapy alone. TMB does not seem to identify responders from non responders either for the combination treatment or chemotherapy alone. [60,61,62,63].
KEYNOTE-189 (NCT02578680)	Exploratory analysis of a randomised controlled trial phase III	616 (48% from the all sample)				tTMB ≥ 175: OS was improved with pembrolizumab + chemotherapy over chemotherapy (HR 0.64; CI 0.38-1.07), PFS (HR 0.32; CI 0.21-0´51)	
KEYNOTE-407 (NCT02775435)	Exploratory analysis of a randomised controlled phase III study	559 (56% from the all sample)	Stage IV squamous NSCLC			tTMB ≥ 175: OS was improved with pembrolizumab + chemotherapy over chemotherapy (HR 0.74; CI 0.50-1.08), PFS (HR 0.57; CI 0.41-0.81)	
CHECKMATE-026 (NCT02041533)	Exploratory analysis of randomised phase III study	312 (58% of the patients who had undergone randomization)	Stage IV or recurrent (PD-L1)–positive NSCLC	Assess the effect of the TMB on outcomes with nivolumab vs. docetaxel	TMB determined in tumor and blood samples by WES 0to100 (low burden) 100to242 (medium) ≥ 243 (high)	tTMB ≥ 243: ORR was higher in the nivolumab group than in the chemotherapy (47% vs. 28%), and PFS was longer (median, 9.7 vs. 5.8 months; HR 0.62; 95% CI, 0.38 to 1.00).	No significant difference was observed in OS between the nivolumab and chemotherapy groups regardless of TMB, according to findings published in the New England Journal of Medicine [33].
CHECKMATE-227 (NCT02477826)	Exploratory analysis of randomised phase III study	679 (58.2% from the all sample)	Stage IV or recurrent NSCLC	Evaluate TMB as a potential predictive biomarker of efficacy of nivolumab, nivolumab + ipilimumab, nivolumab + platinum-doublet chemotherapy and of platinum-doublet.	TMB determined by WES Cutpoint of 10 mutations per megabase	Similar degree of OS benefit in nivolumab + ipilimumab, regardless of TMB (≥10 vs. <10 mutations per megabase, respectively) OS benefit for nivolumab plus ipilimumab vs chemotherapy regardless of TMB or PD-L1	A similar degree of OS benefit was found for nivolumab and ipilimumab regardless of TMB according to findings published in the New England Journal of Medicine Combination of PD-L1 and TMB did not reveal a subgroup with an increased benefit for nivo + ipi vs chemotherapy [64].

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
