# Peer review of "From Tumor Mutational Burden to Blood T Cell Receptor: Looking for the Best Predictive Biomarker in Lung Cancer Treated with Immunotherapy"

_cancers, 2020, doi:10.3390/cancers12102974_

Round 1

Reviewer 1 Report

Dear authors

As we know, tumor immunotherapies are emerging as a beneficial tool for cancer treatment, but resistance during treatment is a major issue. It is urgent to find the befitting biomarkers that predict response to tumor immunotherapies. So there are more and more reviewers and research articles focus on predictive biomarkers for immunotherapy response beyond PD-L1 across multiple cancer types.

This manuscript summarizes recent findings describing the clinical utility of TMB and TCRβ (TCRB) in Lung cancer and concludes that: immune, neoantigen and checkpoint targeted variables are required in combination for accurately identifying patients who most likely will benefit of ICIs. It is well written and useful for readers.

But there are also other published reviews about immunotherapy biomarkers in lung cancer and other cancers recently, for example,

 “The Promises and Challenges of Tumor Mutation Burden as an Immunotherapy Biomarker: A Perspective from the International Association for the Study of Lung Cancer Pathology Committee” which was published on Journal of Thoracic Oncology on June 6, 2020;

“Biomarkers for immune checkpoint inhibition in non–small cell lung cancer (NSCLC)” which was published on Cancer on Nov.6, 2019;

“Characteristics and prognostic significance of profiling the peripheral blood T‐cell receptor repertoire in patients with advanced lung cancer” which was published on International Journal of Cancer on Jan.21, 2019;

“Tumor mutational burden in lung cancer: a systematic literature review” which was published on Oncotarget on Nov.12, 2019;

“Predictive biomarkers for cancer immunotherapy with immune checkpoint inhibitors” which was published on Biomarker Research on Aug.26, 2020;

“Predictive biomarkers of response for immune checkpoint inhibitors in non–small-cell lung cancer” which was published on European Journal of Cancer in January 2019;

So this topic is not new and special. I think you still need to do some major revisions to make this review stand out from many similar reviews.

  1. Although this review is about lung cancer, this manuscript just focuses on lung cancer, rarely describes the immunotherapy biomarkers in other tumor patients. As we know, what's fascinating and special about tumor immunotherapy is that immunotherapies work by using our own immune system to prevent, control, and eliminate cancer, they are not limited to special cancer type, although different cancers are very complicated and heterogeneous. It would be better if this manuscript has a brief description of each immunotherapy biomarkers in all tumor patients, not just lung cancer. Readers may also want to know where lung cancer is consistent and inconsistent with other tumors in immunotherapy. This will give the readers a more complete understanding of the role of immunotherapy in lung cancer.
  2. This review is focused on tumor immunotherapies biomarkers. As we know, for the cancer immunotherapies, safety is as important as efficacy. Serious side effects can affect whether cancer drugs are approved for clinical use. There are a lot of patients who will develop Immune-Related Adverse Events (irAEs) after immunotherapies, some patients have to stop immunotherapy if they have serious irAEs although they may benefit from the efficacy. For example, Tremelimumab, one anti-CTLA-4 antibody, has been undergoing human trials for the treatment of various cancers but has not attained approval for any. A lot of Tremelimumab clinical trials are terminated because of serious toxicity and unsatisfactory effectiveness. This manuscript mainly focuses on the efficacy, rarely describes the safety, although only one short paragraph “A high TMB may be a useful biomarker for assessing patient´s risk of immune-related adverse events (irAES) during anti-PD-1 therapy.” More reviews on irAEs will make this manuscript special and attractive.
  1. Page9, table 2, why there are some blanks for “NCT02041533” and “NCT02477826”?
  2. Page 12, “codon degeneray” or “codon degeneracy”?

Author Response

  1. Thank you very much for your support. From line 48-52 I have made a small summary at the global level about the approval of Pembrolizumab in tumors other than lung cancer. On the other hand, although this review is mainly focused on predictive biomarkers of response to immunotherapy treatment in lung cancer, from line 89-99 of the manuscript, I have done a brief description of each immunotherapy biomarkers in all tumor patients, not just lung cancer. I have introduced several of the recommended references in the article.
  2. As you has mentioned, the safety of immunotherapy treatment is as important as the efficacy. Therefore, following your recommendations in point 3.2 (in red colour) (sorry for the inconvenience but from the tables, the lines of the article are not numbered to be able to easily identify the corrections), I have added a small summary about adverse effects related to immunotherapy in lung cancer and possible molecular biomarkers predictors of immune-mediated toxicity, as well as bibliographic references. As you has said, I find it an interesting and relevant topic right now.
  3. In table 2, in red, I have filled in the blanks referring to NCT02041533” and “NCT02477826” studies. Thank you for your correction.
  4. In page 13, correction done in red: codon degeneracy. Thank you for your correction.

Reviewer 2 Report

This review is well written and provides an overview of the literature regarding TCRvb, TMB and PD-L1 (IHC) labelling as biomarkers that can predict response to ICI. 

Unless I am mistaken, the recent FDA approval (June 2020) is not discussed in this review: : the Food and Drug Administration granted accelerated approval to pembrolizumab for the treatment of adult and pediatric patients with unresectable or metastatic tumor mutational burden-high (TMB-H) [≥10 mutations/megabase (mut/Mb)] solid tumors, as determined by an FDA-approved test, that have progressed following prior treatment and who have no satisfactory alternative treatment options. This was based on a retrospective analysis of 10 cohorts of patients with various previously treated unresectable or metastatic TMB-H solid tumors enrolled in an open-label, multicenter, non-randomized trial, KEYNOTE-158 (NCT02628067). Patients received 200 mg of pembrolizumab intravenously every 3 weeks until toxicity was unacceptable or disease progression was documented. The primary efficacy outcome measures were overall response rate (ORR) and duration of response (DoR) in patients who received at least one dose of pembrolizumab. A total of 102 patients (13% = 102 out of 785 patients initially included) had tumors identified as TMB-H, defined as TMB ≥10 mut/Mb. In these 102 patients 29% (=30 patients) had an ORR (i.e. 30/785 (3.8%) of the patients initially included in the study). Thus for 3.8% of the patients we are moving towards a regulatory obligation to perform a test to determine the TMB  (FoundationOneCDx assay). It seems that it will be expensive and a "gas factory" to only identify 3.8% of patients. What do the authors think? Could they add a comment on that in particular?

Chapter 7 : Correlation TMB-TCRvb; forth paragraph last sentence "Non-smokers with higher TCR diversity than smokers may be more likely to benefit from immunotherapy" ==> I'm not sure that this is clear since the previous sentence says that the diversity of the TCR was greater among smokers. Please clarify this paragraph.

Author Response

1.Thank you very much for your support. I have included what is mentioned below on lines 116-128 in red.

Pembrolizumab received accelerated approval in the United States by the FDA in May 2017 for the treatment of adult and pediatric patients with unresectable or metastatic solid tumors MSI-H / dMMR that progress to standard treatment and have no satisfactory results or alternative treatment options. regardless of the histological subtype or origin of the primary tumor. It was based on data showing an ORR of 39.6% and evidence of lasting clinical benefit in 149 patients with MSI-H / dMMR cancers in 5 clinical studies, including 61 from Cohort A from the KEYNOTE-164 Phase II study and 19 from the KEYNOTE study -158 of pembrolizumab in patients with CRC MSI-H / dMMR and without CRC, respectively. This marked the first approval of an oncology therapy independent of tumor histology or anatomical location of origin, in which treatment is based on a common tumor biomarker.

The recent FDA approval (June 2020) of pembrolizumab for the treatment of adult and pediatric patients with  advanced cancer of any type with a high TMB ≥10 mutations/megabase (mut/Mb)] represents an important therapeutic advance and strengthens the role of genomics in the treatment of cancer.

This approval shows that oncologists must take into account the genomics of the tumor, especially in tumors with few treatment options as occurs in some tumors of children and young adults. Molecular characterization of cancer allows to identify the most effective and least toxic treatment and it is especially useful in tumors without effective treatments. Although we know that not all patients will benefit from this.

Taking into account the study that led to its approval (KEYNOTE-158) (NCT02628067), the final percentage of patients who will benefit from this treatment is small (3'8) and having a tumor with a high TMB is not a guarantee that pembrolizumab will be effective.

MSI-H and high TMB are related because an increase in TMB is also observed in most MSI-H tumors. However, the reverse is not the case: many high TMB tumors do not have MSI-H. The prevalence of tumors with MSI is low (2-4%), however, having a drug approved according to the TMB is important, because it is more common in tumors than MSI-H.

In the KEYNOTE-158 study, almost 30% of the patients who had tumors with high TMB responded to treatment and the responses were long-lasting, also included patients who had progressed to multiple lines of treatment and without further therapeutic options.

Furthermore, tumors in children and young adults do not have many mutations and therefore are not likely to have high TMB. But some of the tumors in children have high TMB, and the approval of this drug means that these young patients now have an important new treatment option based on a molecular biomarker.

2.Thanks again for the correction. This is an error. I have corrected it by mentioning it in chapter 7 about TCR diversity in smokers.

Reviewer 3 Report

This is an interesting and well organized review on the utility of different biomarkers in evaluating the response to checkpoint inhibitor therapies in lung cancer.

Before the review is ready for publication, the authors should review and correct the English syntax, spelling, and subject-verb agreements. Also, there are too many paragraph breaks!

Other minor issues that should be fixed are the following:

In paragraph 4.1, the authors state that “tTMB requires a specific preparation with formaldehyde fixation”. There is no mention of this specific preparation in the quoted reference 19. I think this sentence should be rephrased, because formalin fixation is the typical technique for clinical tissue preservation, but it is not required for TMB. Conversely, it can create artifacts for sequencing due to crosslinking.

At the end of paragraph 4.1, the authors state that “Merkel-cell carcinoma and some virus-induced tumors suggest that the presentation of viral antigens on certain tumor types may confer an increased response rate to ICIs”. However, the quoted reference 46 is about pulmonary sarcomatoid carcinomas and do not mention any Merkel or virus-induced cancers. The reference should be updated with the correct paper.

Author Response

  1. Thanks for your corrections and recommendations. I have tried to check and correct the syntax, spelling, and subject-verb agreement in English that I have identified in the manuscript. However, if you consider it necessary, a more detailed correction of the manuscript will be made.
  2. Thank you very much for your correction. I have modified what is mentioned in the paragraph 4.1. As you said, formalin fixation is the typical technique for clinical tissue preservation, but it is not required for TMB.
  3. Thanks again for your correction. I have corrected the bibliographic reference related to “Merkel-cell carcinoma and some virus-induced tumors suggest that the presentation of viral antigens on certain tumor types may confer an increased response rate to ICIs” at the end of paragraph 4.1. The correct bibliographic reference is number 58 about “Development of tumor mutation burden as an immunotherapy biomarker: Utility for the oncology clinic. Ann Oncol. 2019;30(1):44–56. doi: 10.1093/annonc/mdy495”.
